# COVID-19 in Pluralea Interactions: A Case Study in Santo Tomé (Corrientes, Argentina)

**DOI:** 10.3390/pathogens12020291

**Published:** 2023-02-09

**Authors:** Andrea Mastrangelo, Andrea Alegre, Karina Giménez

**Affiliations:** 1Consejo Nacional de Investigaciones Científicas y Técnicas, Centro Nacional de Investigación y Diagnóstico en Endemo-Epidemias, ANLIS, Ministerio de Salud, Escuela IDAES, Universidad Nacional de San Martín, Buenos Aires C1063, Argentina; 2Instituto Universitario de Ciencias de la Salud, Fundación H.A. Barceló, Santo Tomé W3340, Argentina; 3Instituto Nacional de Tecnología Agropecuaria, Agencia Extensión Rural Santo Tomé, Santo Tomé W3340, Argentina

**Keywords:** SARS-CoV-2, COVID-19, social inequity, social anthropology, environmental health

## Abstract

This paper is an ethnographic case study of COVID-19 emergence in Santo Tomé (South America, NE Argentina, ≂25,000 inhabitants). Based on interviews with healthcare personnel, we describe local containment and prevention policies in a context of national lockdown measures. We reconstruct a tree diagram of infections, index cases and close contacts that spread infection locally. In parallel, fieldwork in a sample of impoverished subsistence agricultures and fishermen allows us to describe drought and fresh food production decline during confinement as convergent ecocrises (pluralea interactions) with the SARS-CoV-2 outbreak. The core idea of the article, which emerged from ethnographic fieldwork evidence, is that in the context of climate change, the sudden onset of an infectious disease interacts with convergent ecocrises.

## 1. Introduction

COVID-19 is the disease caused by the new coronavirus SARS-CoV-2. This disease began in the city of Wuhan, Hubei Province, People’s Republic of China, in December 2019 and was declared a pandemic by the World Health Organization on 11 March 2020. Between 20 March 2020 and 31 December 2021, 287,899,658 confirmed cases accumulated worldwide, with 36% of them concentrated in the Americas [1]. Within this region and period, the predominant variant was Delta. Argentina was the second country, after Brazil, in absolute values of positive cases [2]. However, Argentina had a cumulative incidence rate (22.2 k per 100,000 people) higher than Brazil’s (17.2 k), and the crude fatality ratio—CFR—in Argentina in this period was 1.3%, lower than Brazil’s (1.9%) [3]. The CFR in Argentina is the 4th lowest of the 10 countries in the region, a fact that could be related to the general prevention and care measures adopted by the federal government during that time. Between March and November 2020, public policies in Argentina [1] were given the name of *aislamiento social preventivo obligatorio* (ASPO)—mandatory preventive social isolation. ASPO was centered around respiratory hygiene, handwashing, isolation of those infected and lockdown for healthy individuals, as well as restricting public circulation to food and fuel provision and law enforcement [4].

In this context, we monitored these prevention measures in three different-sized cities (Buenos Aires Metropolitan Area 41 ≈ 14,000,000 inhabitants; Greater Resistencia Metropolitan Area ≈ 400,000 inhabitants; and Santo Tomé ≈ 25,000 inhabitants), in neighborhoods with an average index of unmet basic needs—UBN—(*necesidades básicas insatisfechas*—NBI) equal to or greater than 4%, calculated based on the Argentine census regarding housing (NBI 1), living conditions (NBI 2) and critical overcrowding (NBI 3) [5]. Our underlying understanding was that these structural deficiencies limited the efficacy of the health measures recommended before vaccination.

At the onset of the pandemic, between March and June 2020, the largest share of infections occurred in the first two cities [6,7], but a few months later, epidemic outbreaks (called “second, third, fourth waves”) appeared throughout the country. In January 2021, it was Santo Tomé’s turn. 

This article is a case study of the COVID-19 outbreak in Santo Tomé. Santo Tomé is a rural area with an urban administrative center that includes a university.

The aim of this paper is to study how the epidemic unfolded in Santo Tomé, how the health response was organized and what everyday life was like for residents during confinement. This will allow us to develop an ethnographic comprehension of the entwined disruptive environmental events linked to production of commodities [8,9], scarcity, social inequality [10] and structural violence [11], which are associated with public health securitization [12,13]. Although climate change is not only a natural phenomenon, it is interlinked and interconnected with COVID-19 emergency overlapping crises that should be analyzed with a broad intersectional approach [14]. This description of drought (worsened by hydroelectric dams and forestry water consumption) and fresh food production decline (affected by agribusiness promotional policies) as convergent ecocrises in pluralea interactions [8] shows how modes of economic production determine production modes of disease, risk and death [7,9,15].

## 2. Materials and Methods

This is a case study in a second-level administrative unit (*departamento*, similar to a county) in the Argentine Province of Corrientes, northeast of Argentina, Santo Tomé. It has a surface area of 7094 km^2^ with 61,297 inhabitants as well as a rural–urban municipality that is also named Santo Tomé, where the local government is settled, with 25,824 inhabitants [5] (see Figure 1).

Between March 2020 and December 2021, the anthropologist involved in this study conducted ethnographic fieldwork in Santo Tomé. At the beginning, due to the COVID-19 confinement policy, interviews were conducted by cell phone and, since November 2021, using face-to-face fieldwork. We characterized the local containment and prevention strategies during the COVID-19 epidemic with healthcare personnel and public statistics [5,16,17]. In parallel, we gathered a sample composed of people with unmet basic need (NBI 1, 2, 3, ≥4% or equal to or higher than an average of 4% [5]). This sample comprised a list of small-scale fresh food producers registered at the National Institute of Agrarian Technology—INTA—provided by one of the authors (Karina Giménez), a technician at this institution. Subsistence activities of this rural poor population were described by conducting interviews and participant observation. The sample was referenced in a map, and overlapped with unmet basic needs and forestry. The representative of the main source of local employment, the business association of sawmills, was also interviewed. The face-to-face contact and interviews with businessmen and public health personnel were conducted at their workplaces. 

In all cases, these were open interviews that started with the question “What about COVID-19 in Santo Tomé?” Interview transcriptions, written texts provided by informants, sampling of georeferenced points and photographs taken as part of the anthropologist’s fieldwork were collected in a folder named Fieldnotes. This evidence is referenced in this text as (Fieldnotes, November–December 2021). 

The convergent ecocrises (drought and downspout of the Uruguay river) during the COVID-19 outbreak were described through secondary sources [18,19,20,21,22,23,24,25,26,27]. 

## 3. Results

The results are presented in two sections. Section 3.1 describes the local public health response to the COVID-19 emergency, including a contagion tree diagram reconstructed by health personnel during ethnographic fieldwork. Section 3.2 analyzes the convergent ecocrises (drought, downspouts and the local fresh food production decline) with the emergency of COVID-19.

### 3.1. COVID-19 in Santo Tomé

The public health approach to the COVID-19 emergency in Corrientes was centralized. To preserve the local capacity for care for acute and chronic patients beyond COVID-19, hospitals throughout the province restricted themselves to diagnostic and ambulatory care, while limiting hospitalization to COVID-19 cases. The different cities and towns, within care bubbles, carried out only the diagnosis of new infections (taking “swabs” or samples for antigen testing) and tests for follow-up after infection (X-rays and CT scan). The diagnostic confirmation using PCR, as well as the care of cases with poor clinical evolution, required referral to the hospital in the capital city of the province, 390 km away from Santo Tomé. Recovered and dead individuals would, then, return to their places of origin.

Vaccination started in September 2020, and was also managed centrally. Those eligible could register online and, when vaccines became available, they could go to the regional hospital to be vaccinated.

From the perspective of Santo Tomé’s health system agents, the centralization of acute care was efficient, but in the case of vaccines, it was disorganized and discouraged vaccination. There were people who were notified that it was their turn to get vaccinated, only to find upon arrival that the vaccines had not been sent to the hospital. Others did not understand the process of making an appointment and went to the hospital without one. These situations caused rumors stating that getting the vaccine “was a mess and people were fighting in the entrance of the hospital.” (Fieldnotes, November–December 2021).

#### 3.1.1. About COVID-19 and Lockdown in Santo Tomé, Corrientes

Between 17 March 2020 and 19 September 2020, the inhabitants of Santo Tomé, Corrientes watched TV and stayed home, complying with the lockdown established for the prevention of community circulation of SARS-CoV-2.

Santo Tomé is a city with an international border via the bridge to São Borja, Rio Grande do Sul, Brazil. New, large-sized trucks (Globetrotter brand) enter through this bridge, and due to both the transport companies’ organization of work and the length of the route (3327 km), drivers change to continue along the Mercovía transnational roadway. The transit of land vehicles includes trucks transporting cars that come from the Brazilian automotive industry and are imported to Argentina or Chile. In addition to international commerce, 90% of food products are transported via these means of transportation. There are 64 truck drivers living in Santo Tomé.

The index case was not a Santo Tomé truck driver, but actually the girlfriend of a truck driver, who had accompanied him from Posadas (Figure 1) on a return trip. They arrived in less than two hours and, the next day, she took a bus back to Posadas. In the interprovincial border between Corrientes and Misiones, she took a rapid antibody test and tested positive. They told her she could isolate in a collective isolation center set up by the province of Misiones in the *Crucero del Norte* soccer stadium. She refused and took a bus back to Santo Tomé, where she told her boyfriend, who also tested positive and remained isolated. However, since she was not tested by Corrientes’ local health system, she kept circulating throughout the town and running errands, as she was asymptomatic. It is based on this case that town officials and residents narrate the onset of community circulation.

#### 3.1.2. The Santo Tomé Crisis Committee

Santo Tomé has had a Crisis Committee for the prevention of infectious diseases since the emergence of leishmaniasis and the recurrent dengue epidemics, and they also convened to manage the emergence of COVID-19. Participants of the committee include the mayor of the municipality, the directors of *San Juan Bautista* Hospital, the primary care director of the hospital, the head of Health Region No. 5 of Corrientes province, the academic secretary of the School of Medicine and a professor at a private university with a local branch. Security forces (gendarmerie, prefecture, firefighters and the police of the province) also participated. The committee members acknowledged the mayor as the maximum authority.

The COVID-19 emergency committee in Santo Tomé was organized into four medical areas:(1)Call center: Provided instructions on isolation for positive cases, cohabitants and close contacts and prepared public statements. Coordinated by two doctors, with calls answered by local students in their seventh year of medical school.(2)Monitoring and notification: Three clinical doctors made daily phone calls to positive cases for ten days after the diagnostic confirmation. According to the protocol, 45-year-olds and over had a TC scan on the fifth day after confirmation. People under 45 had a routine x-ray and a TC scan if, after 10 days, their condition had worsened.(3)Primary care and vaccination were coordinated by a doctor.(4)Nasal swabs and testing were coordinated by a biochemist.

Given the situation in which the index case emerged, the committee decided to create a collective isolation center. The truck driver and his girlfriend lived with six other people, three of whom belonged to at-risk groups. The center was created in a commercial property downtown, with proper hygienic conditions, and those infected were transported there. Five days later, municipal authorities were informed of a criminal complaint of unlawful deprivation of liberty and the controlled isolation facility was closed.

The isolation of the truck driver and his girlfriend in an ad hoc institution was the only such incident and it was considered a failure; thus, all control of community circulation from September 2020 until the time of the fieldwork (end of November 2021) took place at people’s homes. Between April 2020 and April 2022, Santo Tomé had 5340 total cases of COVID-19. Santo Tomé cases of COVID-19 constitute 4% of the total number of infected individuals and 4.35% of deaths in the province during that period. The province of Corrientes contributed 1.5% of the total number of infected individuals in the whole country, with a crude fatality ratio equal to the national value of 1.3% [16]. 

An analysis of the 50 press releases from the Santo Tomé Crisis Committee that were issued between March 2020 and April 2021 allowed us to identify that securitization of public health was essential to regulating the circulation of the population to control community spread of the virus. In this context, some of the health surveillance actions fell to public security forces, and others were judicialized (lack of compliance with isolation, seizure of personal vehicles). This way, punitive social control adapted its repressiveness in order to encourage compliance with biosafety norms [12,13]. In Santo Tomé, control was centered on nighttime activities, as well as individual pedestrian circulation or multiple people’s presence vehicles in which only the driver could be present. Similarly, the Naval Prefecture ensured that the river border could not be crossed, while the National Gendarmerie monitored the absence of symptoms and negative tests in cargo drivers at the international border, and the local police did the same for those entering the city. 

As we highlighted earlier, the local index case occurred in September 2020, but community circulation remained low until April 2021, when the “Santo Tomé peak with the Delta variant” took place (Figure 2).

Based on fieldwork interviews with doctors in charge of the call center, we identified three major foci of contagion in a tree diagram: truck drivers, families from the *Centenario* neighborhood and sawmills. 

#### 3.1.3. Truck Drivers and COVID-19

Since the first meetings in March 2020, the Santo Tomé Crisis Committee took a protocolized approach to truck drivers who lived in the locality. Starting on 27 March 2020 it was established that all truck drivers from international transportation companies residing in Santo Tomé and whose activity was exempted in the National Decrees had to contact the Crisis Committee from 9 AM to 7 PM to notify that they were leaving town, with the objective of coordinating their return to Santo Tomé. Those who did not comply with this would be unable to reenter the city upon their return. (Crisis Committee Press Release, 27 March 2022).

The efficiency of this control measure varied, as many truck drivers, although they respected the isolation, returned without symptoms and were not tested. Days later, symptoms would appear in members of their families. The result of this experience and the failure of isolation in collective institutions led to the following observation from a doctor of the committee:

“Families of truck drivers cannot be isolated *ad infinitum* –permanently-, and the drivers, as essential workers, go on long interjurisdictional trips and return to their homes. If the drivers present false negatives or they are asymptomatic and don’t voluntarily get tested, their families can’t remain isolated because often they are women taking care of young children and they have to stay active and circulating to keep their houses stocked, or because they carry out essential activities in the locality related to cleaning, health or food supply.” (Fieldnotes, November–December 2021).

#### 3.1.4. COVID-19 at Centenario Neighborhood

The topography underlying Santo Tomé is composed of small patches of wood saw (*monte*) interspersed with floodlands (*bañado*, low-lying areas with grass that retains water). A walk around *Barrio Centenario* allows one to infer that the area is a floodland. The depressed topography and the problems with sewage drainage of surface waters make Centenario a poor neighborhood located near the city center. It is a neighborhood with precarious houses, saturated drainage ditches and overcrowding, with many caregivers living there who continued to work during confinement. There were many families with COVID-19 patients in this neighborhood, as there were up to 10 people living together in many of the homes. One by one, they tested positive until all had caught the disease, meaning that the last to contract COVID-19 had spent over 50 days in isolation.

#### 3.1.5. Sawmills and COVID-19

Sawmills were able to gain exemption from lockdown. In March, when the stoppage of all activities was decreed, the owners of sawmills avoided the restriction until early April. During April, they instituted shorter workdays. In May, after negotiating and presenting a health protocol, they were able to resume regular working hours. National and international demands began to grow. Cargo transport was not restricted, and river transport used containers that allowed them to export. The second semester of 2020 was satisfactorily profitable.

On average, between March 2020 and December 2021, 30% of the staff in each sawmill was infected. Given that work is carried out on a production line, this reduced the overall productivity of the plants. The impact was even greater considering the 14-day isolation periods that close contacts had to observe. In no case did the workplace insurance cover the cost of medical leave or preventive isolation for workers. The abundance of cases and economic losses in the sector led to massive testing units in the sawmills being installed when the winter of 2021 started.

### 3.2. Lack of Water, Subsistence Agriculture and Artisanal Fishery during COVID-19 Lockdown

The largest part of the population with NBI 1, 2, 3 and ≥4% [5] is found in the peri-urban area of Santo Tomé (Figure 2), and they practice subsistence agriculture with commercialization of the small surplus. The analytical category of subsistence farming in the northeast of Argentina, in the context of late capitalism, identifies productive units corresponding to three agrarian social types: a majority of subsistence agriculture families of peasants with customary land tenure; farmers impoverished by customary land tenure, loss of capital and markets of yerba mate and tobacco (Table 1); and semi-proletarian landless rural workers selling their labor force seasonally [28]. In this Santo Tomé case study, semi-proletarian landless workers are called artisanal fishermen.

During fieldwork, in November 2021, at least two peaks of COVID-19 had taken place in the area. However, the most challenging issue in the town was the drought, although the municipality of Santo Tomé and the urban and suburban areas are close to large volumes of fresh water. The bordering Uruguay River forms part of the international watershed of the Río de la Plata, traversing Brazil, Uruguay and Argentina (1779 km long and a historic average flow rate of 4622 m^3^/s). Just 111 km away are the Esteros del Iberá, a wetland ecosystem with an extensive complex of bodies of water that covers 12,000 km^2^. This ecosystem is mainly connected to the Paraná basin [24] to the west of the study area. 

With respect to subterraneous waters, the Guaraní Aquifer System, one of the largest freshwater reserves in the world, is found below Santo Tomé. It is located between 16° and 32° S parallels and 47° and 60° W meridians, occupying an area of around 1,100,000 km^2^. Its area overlaps with part of the Río de la Plata Basin [15].

The present-day climate in the region of the study is humid or sub-humid (1200 to 1500 mm of annual rainfall). Nevertheless, 2021 saw the lowest rainfall in the last 90 years, with a deficit of 600 mm/year. This drought is not predicted to improve during 2022 [19,21,23].

For the wetlands, river and aquifer to recover their historical water levels, first it is necessary that the soil becomes saturated, which, in current conditions, could take between two and three years. During the fieldwork, the drought was the first topic of conversation with the interviewees. Many producers did not have access to tap water, and in rural areas, either the wells and pumps were not bringing up enough water or the electric pumps had burned out from overuse. Wells dug to water crops had to be used for human consumption. 

#### 3.2.1. Lack of Water and Niña Cycle

The droughts in the Argentine Littoral and Pampa Regions are cyclical. Meteorological research studies have determined that the oscillating heat and cooling pattern of the tropical and subtropical Pacific regions, known as the El Niño–southern oscillation (ENSO) cycle, directly affects the distribution of precipitation in tropical areas and can have a strong influence on the climate in other parts of the world. El Niño and La Niña are extreme phases of the ENSO cycle; between the two there exists a neutral phase. In the Río de la Plata Basin, El Niño (heating) is associated with an abundance of rain, and the cooling of La Niña with droughts. The Niño/Niña cycles alternate every three to seven years. Meteorological records identified that the current Niña episode began in the second semester of 2021 and, by March 2022, there was a 65% chance that it would continue all year round [20]. However, as the drought in Santo Tomé began previously, in 2020, water shortage due to La Niña partially explains Santo Tomé’s deficit. The rainfall deficit registered in the northern area of the hydrological region during the period of January–November 2021 was the 14th driest in the 1961–2021 period, with a rainfall deficit of −302.8 mm [21].

#### 3.2.2. Lack of Water, Forest Industry and Hydroelectric Energy 

Although, meteorologically, this drought was not the worst on record since 1961, locally, in ST, there are abundant testimonies to the fact that no one remembered anything like it. No one understood why there was no water.

The river level records in 2021 show peaks of shallow water levels, under 2 m, in March and November 2020 and April–May 2021 [19]. In December 2021, the Naval Prefecture published warnings to not engage in water sports given the danger of harm caused by rocks [22].

As we have mentioned, this was not the first La Niña cycle experienced by the residents of Santo Tomé, but it was the first low water level accompanied by drought seen since Brazil put into effect three hydroelectric dams upstream—Foz de Chapecó (855 MWts, inaugurated in 2001), Itá Hydroelectric Power Plant (1450 MWts, also inaugurated in 2001) and Machadinho (1140 MWts, inaugurated in 2002)—that require water to be retained in order to keep their costly turbine generators working.

Regarding the generation of electricity, locally, the “sustainable alternative” is a biomass plant that uses residual sawdust as fuel. The first power station, which produces 40 MWts, is located 43 km away from the urban area of Santo Tomé in another locality of the Santo Tomé department, Gobernador Virasoro (Figure 1), and there has been an environmental impact report approved by the Correntinean Institute of Water and Environment (*Instituto Correntino del Agua y el Ambiente*—ICAA) for a similar plant, Santo Tomé Kuera, in the locality of Santo Tomé [29].

Sawdust transformed into energy comes from the implanted *Pinus* spp. and *Eucaliptus grandis* forests in the whole region, especially from nearby sawmills, 12 of which are located in the suburban area of Santo Tomé. 

In 2002, the first year in which the three dams were operative, Corrientes had 283,028 forested hectares (Table 1) and a sustained development plan to grow in that area, which is now coming to fruition. In 2011, it was confirmed that there were 300,000 forested hectares [24], and in 2022, the Ministry of Production hoped to reach 500,000. Since the late 1990s, lumber production has been promoted nationally, with fiscal retributions for sowers (Ley Nacional 25.080). This tax promotion and the lack of incentives for tobacco and yerba mate explain why small and medium-sized production units predominate in these historical harvests, with 5.4% of land use (Table 1).

As shown in Table 1, lumber plantations are concentrated in the hands of large-scale producers, who possess 96.6% of all the land being produced in the province. Corrientes is part of the Mesopotamian region and concentrates 76% of the forested surface area in the country, with the province making up 30% of the national total. Around 50% of the 300,000 hectares of planted area in Corrientes in 2011 was located in the departments of Santo Tomé and Paso de los Libres (along the coast of the Uruguay River), and 75% of the implanted forest was made up of *Pino ellioti* [30]. In 2015, the planted surface area of *Pinus* spp. tripled that of eucalyptus [27].

According to the President of the Association of Wood and Related Industries (*Asociación de Industriales de la Madera y Afines*) from Santo Tomé, in 2021, the department possessed 12% of the forested area of the country and, taking into account the number of sawmills, it is the primary industrial lumber district in the province. Given the centrality of lumber activities, the primary economic activity in the municipality is forestry. The labor force is composed of forestry maintenance workers coming from outside the locality and local sawmill workers. There are an average of 50 workers per sawmill, with an estimate of 700 total workers. Investors in sawmills are from outside the locality. A small number of unsawed trunks are taken by train to other preparation centers.

Neither of the two planted pine species (*taeda or elliottii*) are autochthonous; both are North American in origin and can grow well in poor-quality soils, such as lowlands and floodlands, where *Eucaliptus grandis* does not grow. Another characteristic that sets these species apart, especially *elliottii*, is the production of resin. Since 2011, in the locality of Ituzaingó, there has been a plant that produces the sap of pine resin from which rosin (used to make the bows of musical instruments) and solvent turpentine (used to dilute oil paints) are derived [31].

There are other industrial chains at the provincial, regional (Misiones, Entre Ríos and Buenos Aires) and international (Republic of Uruguay) levels: 51% of tree production in Corrientes is used for paper pulp, 27% goes to sawmills for lumber and packaging and 22% goes to boards such as fiberboard, MDF and OSB [31].

The productivity of resin, cellulose fiber and lumber of a tree plantation is associated with genetic characteristics, silvicultural management and the type of soil, but mainly with the availability of water. Water consumption of pine plantations has been studied both in tropical forests [29] and in the Argentine province of Misiones for the *taeda* variety [30]. In the study from Misiones, located north of the study area, water consumption was analyzed through the heat dissipation technique [32,33,34] in three types of silvicultural implantations: forest screen implantation, with more than 1100 trees/hectare; forestry implantation, with less than 1100 trees/hectare; and silvopastoral systems, with up to 300 trees/hectare. In forestry implantation, an estimated daily consumption of 60 liters/tree for a tree with a 40 cm diameter at breast height (DBH) was calculated [31]. Every m^3^ of wood produced consumes around 1.5–2 mm of water [32,33,34]. These levels of water consumption in plantations are not considered alarming given the average rainfall predicted in the region (1800 mm/year). This estimation and the lack of concern clearly does not take into account either the Niña phases of extreme drought in the ENSO cycles or the shortage in the basin regulated by foreign hydroelectric dams.

At the time of the fieldwork (2020–2021) and the writing of this article in 2022, the National Water Institute established that the flowrate of the Uruguay River was at its lowest level in history [23] and that the drought worsened with the forest fires surrounding the urban area in February 2022 [35]. 

Both types of energy production (hydroelectric and biomass) are related to drought. On the one hand, the actual hydroelectric option regulates the low levels of the river to its benefit. On the other, the “sustainable” option consists of generating energy based on biomass due to the consumption of water from tree plantations. Thus, both in the present and in the future, Santo Tomé will “stay with the trouble”, just as Donna Haraway [36] states in the name of her book.

#### 3.2.3. River Waters and Subsistence: Artisanal Fishery

The Uruguay River in Corrientes provides water for birds, as well as abundant vegetation and food in the form of fish bred in its waters. Like other rivers of the South American Neotropical region, it contains abundant and diverse fish; in the Uruguay River, 81 species have been identified, half of which have recognizable ecological or social utility. The dominant families are *Characidae* (23%), *Loricariidae* (20%) and *Cichlidae* (11%), with a species composition similar to that of the Paraná River [37].

Other highly valued species include *dorado* or *pirayú* (*Salimuns brasiliensis*). Fishing of this species has been banned in certain seasons for almost a decade, usually between September and December/January to respect the reproductive season, as occurred in 2020. However, in July 2021, more extreme measures were announced that were connected to the drought, low river level and diminishing of some species [22,25,26]. 

During the fieldwork, we began our survey of resident fishermen in Santo Tomé who sold their catch locally by interviewing the President and the Treasurer of the Santo Tomé Trinity Association of Fishermen (*Asociación Trinidad de Pescadores*, legal status 139/2013). These fishermen define themselves as “artisanal” as compared to commercial fisheries, their Brazilian antagonists, who are more capitalized. Brazilian fishermen, as described by their peers from Santo Tomé, are businessmen who always use boats with motors and fish trawling at night. The Santo Tomé fishermen have pine or *timbó* (*Enterolobium contortisiliquum*) canoes with oars—preferably made of *timbó*, *loro* (*Bastardiopsis densiflora*) or chinaberry wood, able to stand the force of the water—and small 5 hp engines. It is estimated that active fishermen can collect approximately 100 kg/day, selling their catch at some ARS 600/kg (USD 3/kg). This 100 kg daily catch is made up of a number of species, such as *boga, patí, pacú, armado*, and—in the right season and when the water level is high—*surubí* and *dorado*. The productivity of the river is fairly constant from February to December, although limited by bans and low water levels, which change the sailing conditions and the availability of species. This makes fishing a non-fixed contribution to the family income.

The Santo Tomé fishermen describe the commercial fisheries “of Brazilians” as predatory, as they use larger boats and trawls, as well as night labor. Artisanal fishermen are subsistence fishermen who sell 50% of their catch and often also carry out agricultural activities with their families. They are obligated to respect the ban, but they can always catch scaleless fish (such as *catfish*, *patí*, *armado* and *surubí*) to eat (but not to sell), even during the ban. The fishermen formed a union to formalize the demand to the Naval Prefecture for the right to fish for feeding purposes. Regulations must be complied with when the river level is between 4 and 5 m, but in 2020–2021, it was at 2.35 m and still receding. The artisanal fishermen considered that, given the drop in their income due to the lockdown, they were not in a condition to accept the fishing ban. 

During the lockdown, the Naval Prefecture included fishing as a restricted activity, especially as it is possible to cross the river border to Brazil, and it was necessary to control contact as Brazil scarcely limited circulation. The artisanal fishermen showed the scars on their fingers from cuts received while bringing in the trotlines. Their identification as “artisanal” fishermen based on the vehicle and fishing technique has to do with the fact that trawling (with woven nets) is pursued by the river authority (Argentine Naval Prefecture) that issues sailing licenses. In this sense, the Association and the Prefecture work together to control the regulations. Security forces ask for the ID card of the association members, as well as the official registration of nautical drivers. It is also with this federal authority that they negotiate the permission to fish during bans when there are no other options for subsistence.

From March to September 2020, due to the ban and lockdown, fishermen were stopped for six months. “We became vegetarians,” they say ironically; “the river was so low that there wasn’t any groundwater, we watered the garden with pitchers.”

People lined up in the *Barrio Cerro* neighborhood, where there was a large water tank. During November 2021, the fishermen interviewed stated that in January 2021, people with cars waited in line and loaded them up with water for their farms. At that time, there was so little water pressure in the water supply network of the private company Aguas de Corrientes for the whole city, that the resistance coils burned out in electric water heaters.

As there was an electoral campaign going on in 2021, the fishermen’s association received food assistance from one of the mayoral candidates almost every month. They made community soup pots, but not just for fishermen. The association made such soup pots four times, twice in *Barrio Cerro* and twice in *Puerto Torre*.

Given the low river level and the lack of fish from the species most desired by consumers (*surubí*, *dorado*, *armado*, etc.), the alternative that fishing families found to add value to their work was to make dough-wrapped fish snacks known as *empanadas* using smaller, spinier fish (with less meat and more bones). These species were not banned, and could be caught for self-feeding and selling even with the river at low levels. They are scaled species, the best known of which is *boga*. 

#### 3.2.4. Irrigation Water for Small-Scale Food Producers

Nowadays in Santo Tomé, agrarian food producers are smallholders with up to 25 hectares, and they represent 22% of farming exploitation while possessing 0.2% of the land for agricultural use [17]. Since the 1990s, the agrarian economy from the northeast region of Argentina has been dominated by the forestry industry, with large investments that led to concentrated land ownership. As shown in Map 2, the majority of these producers are located in the rural peri-urban area in census tracts with average NBIs of 1, 2, 3 and ≥4%. Their situation is marginal, but they are part of the rural poor population living in the peri-urban area, so the land demand for lumber activity is not without tension. To strengthen the presence of small-scale producers in the local economy, in 2006, the National Secretariat of Family Agriculture generated wells and pumping systems that made it possible for the children and grandchildren of the original producers to repopulate productive farms. Policies to strengthen family agriculture supported the *Feria Franca* (a local commercialization market for small-scale producers of fresh food) in the construction of the current facilities, finished in 2015 without a bathroom or a kitchen. The annexed bathrooms and kitchen were constructed under the mayor’s administration in 2021. The refrigerated display case and the scale were provided through the federal government’s advice, and seeds were delivered by another governmental program named *ProHuerta*. A cooler for fresh food transportation was provided by the mayor (2015–2019).

In our study, from a list of 25 productive units provided by the National Institute of Agropecuarian Technology, Rural Extension Agency Santo Tomé (*INTA, AER Santo Tomé*), we were able to locate and characterize 20 as active. The workers of the market *Feria Franca* are generally the female partners of male small-scale livestock producers (Figure 3).

## 4. Discussion

Ethnographic fieldwork evidenced convergent ecocrises (pluralea interactions) around forestry agribusiness as a local form of global climatic change that amplifies, compounds and creates new forms of environmental injustices and stresses. As other authors have explained [8,14], climate change is not exclusively a fact of nature. Sultana [14] pointed out that the ongoing climate change amplifies, compounds and creates new forms of injustices and stress, all of which are interlinked with the emergence of the COVID-19 pandemic. It has co-created new challenges, vulnerabilities and burdens, as well as reinforcing old ones. Our field research documented how large hydroelectric dams, forest industry water consumption and fires overlapped with natural phenomena such as the Niña cycle, Uruguay river downspout and COVID-19 emergency. Local modifications to the landscape by forestry implantation or fires, job opportunities in sawmills and fresh-food supply all influenced the risk of becoming sick with COVID-19 in Santo Tomé. The COVID-19 outbreak in Santo Tomé appears in the context of the convergent ecocrises of drought and the decline in local fresh food production.

On the other hand, Sultana [14] referred to the fact that the connections between climate breakdown and the COVID-19 pandemic have exposed the underbelly of structural inequities and systemic marginalization across scales and sites. In our article, we demonstrate how fresh-food supply in Santo Tomé is based on smallholder producers with uncertain land tenure and lack of water in a context of impoverishment caused by traditional harvest decline and agribusiness intensification. The COVID-19 health emergency highlights dilemmas faced by food supply in Santo Tomé. Thus, it can be seen how the predominance of certain productive activities (forest industry) put local food sovereignty in danger. In the case of Santo Tomé, a locality with relative abundance of surface waters and oriented to the lumber industry, a combination of drought, low water levels in the Uruguay River and lockdown revealed a strong dependence on food supply from other localities. In fact, during the COVID-19 confinement, it was the delivery of supplies by truck that introduced the virus into the city. This spread through essential economic exchanges showed that nationwide lockdowns had unintended consequences for other health issues and impoverishment, and they are not a one-size fits-all approach [10]. This points out the need to understand the pandemic from an intersectional, broad health, economic and social perspective [14,38].

As shown by another study in Brazil [39], international trade of food and commodities was a factor in the spread of COVID-19. The supply of fresh food for the cities and international food trade Especially interrupted the lockdown and promoted close contact. As we show in our research, food supply and COVID-19 prevention policies are related in more than one way. Support of family agriculture as a palliative policy for rural poverty has shown good intentions, but has been erratic. The lack of continuity of social agrarian policies during lockdown made the situation of the rural poor population even more critical. As Frempong et al. [10] analyzed for sub-Saharan Africa, the secondary economic effects of lockdown were worse in undeveloped countries. Lockdowns push those who are already poor into extreme poverty and threaten the livelihood of the most vulnerable. In Santo Tomé, all of this happened in a context of severe drought. As in other rural areas of Argentina and Uruguay, subsistence agriculture was displaced by agribusiness expansion [40].

As other authors found in China and Hong Kong [12], we observed in Santo Tomé that restrictions of movement on public roads challenged the protection of citizens’ rights. Nighttime street circulation between rural areas and the city was especially subject to biopolitical control [13]. Even so, our field experiences in Santo Tomé did not lead to the stigmatization of professional or socio-ethnic stereotypes [13], even though this could be documented in larger cities such as Buenos Aires and Resistencia [6,7].

## 5. Conclusions

In Santo Tomé, the public health approach to the epidemic was centralized, and a local crisis committee managed the COVID-19 emergency. The index case was a relative of a fresh-food supply truck driver. After the outbreak of the Delta variant in April 2021, doctors in charge of the local committee identified three major foci of contagion in a tree diagram: truck drivers, families from the Centenario neighborhood and sawmills. 

Truckers who were at risk due to intensive international trade and fresh food delivery were able to break lockdown by traveling long distances. Centenario is an overcrowded urban neighborhood with a low topography, leading to troubles with sewage drainage, where many caregivers and truck drivers reside. Overcrowding promoted close contact. Sawmills are the main employers of local labor and were recognized as a rural activity, so they were able to ignore lockdown movement restrictions.

The emergency of COVID-19 in Santo Tomé converged with a severe drought and a downspout of the Uruguay River. Water in Santo Tomé depends on the hydrological cycle of its rain (1200–15,000 mm/year) refilling the courses of surface water that irrigate plants and allow for fishing. In a socioenvironmental context of drought such as the one studied herein (Niña cycle), as there were hydroelectric dams and agribusinesses planting timber throughout the territory of Santo Tomé with a high water demand, little margin was left for collecting and administering water for consumption and subsistence farming. Map 2 shows how the urban area of Santo Tomé has limited availability of land for vegetable farming, given the intensification of lumber plantations. 

Since 2019, the Uruguay River has been at a historical low and has a high concentration of agrochemicals (cyhalothrin, deltamethrin, bifenthrin, cypermethrin, fenvalerate, endosulfan, permethrin, malathion, triflumuron and chlorpyrifos [26,41]). The confined Guaraní Acquifer System has little availability in the area of study, and the Iberá wetlands, in connection with the Paraná Basin, offers a limited contribution. 

Ethnography has allowed us to visualize how water scarcity led to lack of fresh food (artisanal fishermen became “vegetarians”, growers were without irrigation). Traditional smallholders became impoverished due to the precarious conditions of land ownership, as well as their aging and decapitalization. The economic centrality of the forest industry explains the concentration of land use in forest plantations, the intensity of the drought, the local decline and the external dependence on the production of fresh food. In summary, the agribusiness production mode in Santo Tomé led to the index case of COVID-19, which was in a truck driver (and their close contacts). Likewise, given that the main salaried employment was in sawmills and that this activity was not restricted due to the health emergency, the chain of infections remained active throughout this route. 

On the other hand, restrictions of movement on public roads centered on nighttime activities were the main issue in the press releases of Santo Tomé’s crisis committee. The control of lonely car drivers and pedestrian night prowlers became a relevant issue for biosecurity. In the context of an epidemic, this action is characterized [12,13] as public health securitization. 

## Figures and Tables

**Figure 1 pathogens-12-00291-f001:**
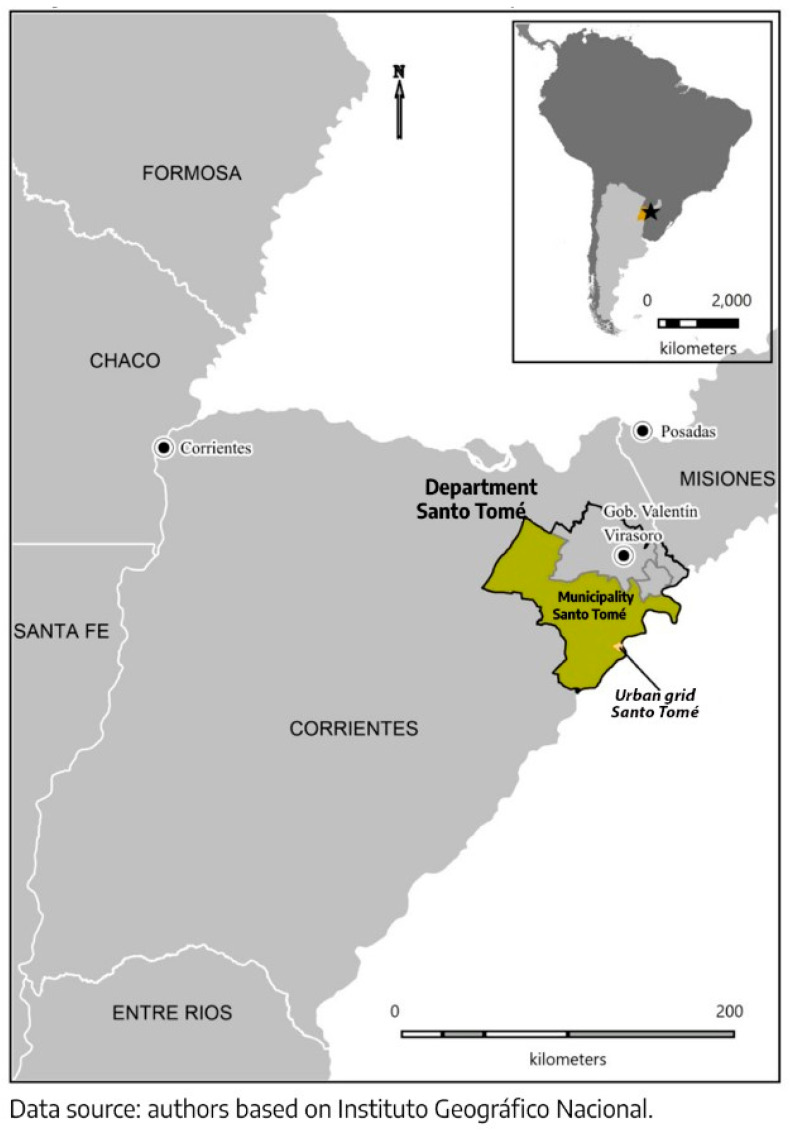
Location of Santo Tomé and nearby towns.

**Figure 2 pathogens-12-00291-f002:**
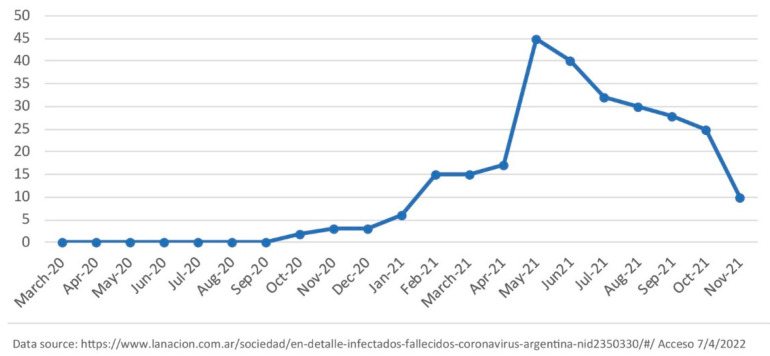
Santo Tomé, Corrientes. COVID-19 average cases in the last 7 days.

**Figure 3 pathogens-12-00291-f003:**
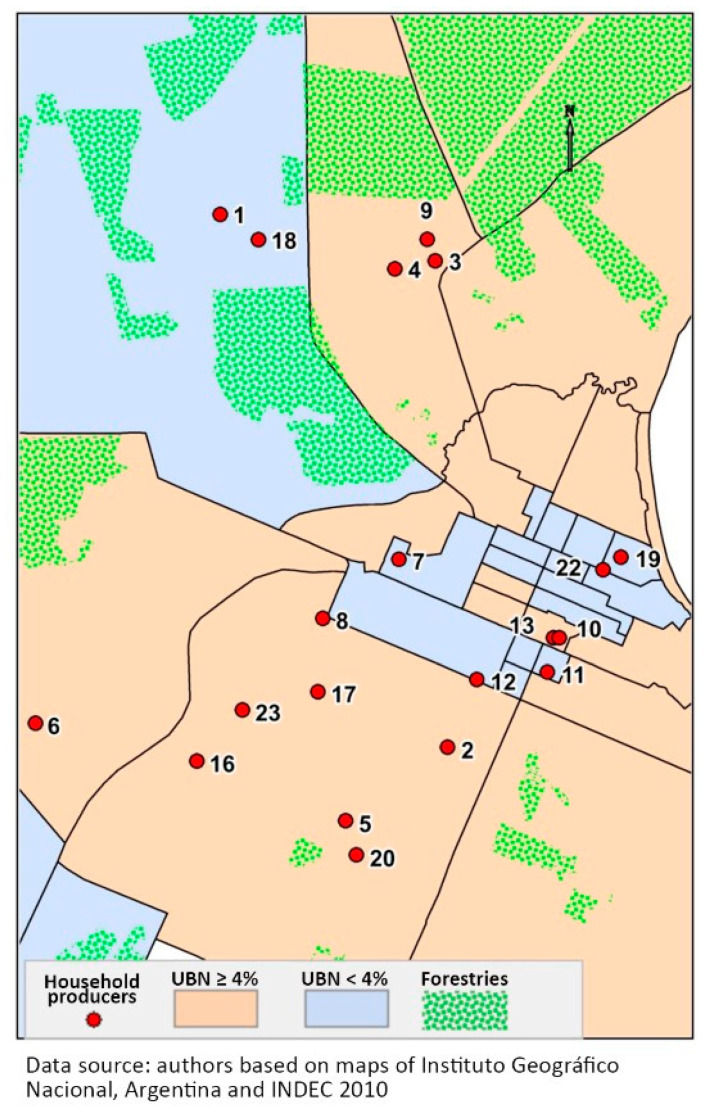
Household producers in areas with unmet basic needs of 1, 2, 3 and ≥4%, as well as forestry areas.

**Table 1 pathogens-12-00291-t001:** Main crops and their occupied areas by type of producer. Corrientes 2002.

Scale Production	Quantity of Units of Production (EAP)	Area in Hectares	Area over Total (%)	Area with Yerba Mate (Hectares)	Area with Yerba Mate (%)	Area with Tobacco (Hectares)	Area with Tobacco (%)	Area of Implanted Forest (Hectares)	Area of Implanted Forest (%)
Small	10,922	611,688	3.8	3913	26.2	3168	88	6114	2.2
Middle	1460	169,003	1.6	1969	13.2	291	8.2	2717	1.2
Big	2855	6,079,573	94.6	9041	60.6	138	3.8	274,197	96.6
Total	15,237	6,860,264	100	14,923	100	3597	100	283,028	100

EAP (Explotaciones Agropecuarias) are units, as counted by the National Agropecuarian Census in Argentina. https://www.indec.gob.ar/ftp/cuadros/economia/cna2018_resultados_definitivos.pdf (accessed on 7 April 2022). **Data source:** authors based on Slutzky 2011.

## Data Availability

Not applicable.

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
