# Peer review of "COVID-19 in Pluralea Interactions: A Case Study in Santo Tomé (Corrientes, Argentina)"

_pathogens, 2023, doi:10.3390/pathogens12020291_

Round 1

Author Response

Answers point by point to Reviewer 1 are attached

Reviewer 2 Report

The article provides a very interesting case study of how the pandemic and other (ecological) crises can interact. However, as it is, the argument is not brought out clearly and it should be embedded in more anthropological literature on the topic (not specific to the case). I included more detailed comments in the attached PDF.

Author Response

Point by point answer to Reviewer 2 is attached

Round 2

Reviewer 1 Report

Dear Authors, 

You manuscript entitled "COVID-19 in pluralea interactions. A case study from Santo Tomé (Corrientes, Argentina)" has been reviewed.

The revised manuscript deserves attention since it represents an ethnographic case study of COVID-19 emergence in Santo Tomé - Argentina.

In general, the manuscript, in its new version, is better than the previous one, but unfortunately in needs a lot of work and modification in almost all its sections (Introduction, Materials and Methods, Results, and Discussion). The English is better compared to the previous version, in addition the references are better presented.

Please find below my remarks and comments concerning your present manuscript:

Minor Queries / Comments:

1- In the whole manuscript, authors are invited to replace the term "Santo Tomé" by "ST".

2- In the Abstract section, Line 62, authors are invited to correct "SARS-CoV-2"  instead of "SARS-CoV2".

3- In the Keywords section, Authors are invited to remove "Infection by coronavirusand to add "SARS-CoV-2" and "COVID-19".

4- In the Introduction section, Line 86, authors are invited to remove (ST).

5- In the Results section, line 231 and line 416, authors are invited to explain what do they mean by the follow:  (Fieldnotes, Nov-Dec 2021).

6- In the Results section, line 410, authors are invited to explain what do they mean by "ad infinitum".

7- In the Results section, Paragraph 3.1.5, is it "sawmills" or "saw mills"??

8- In the Results section, Table 1, the font size is very small, authors are invited to use a larger font size in the cells.

Major Queries / Comments:   

1- Authors are invited to change the type of submitted paper, from Article (which means original research article) present above the title to Case-Study, and then they have to change the anatomy of this paper, as the anatomy of Case-Study.

2- Concerning the Introduction section, this section is very small, it needs to be more rich in information about SARS-CoV-2 and COVID-19 worldwide and in Argentina. In addition some information in the objective must be in the introduction. (Here are some references that can be used in this section: The emergence of SARS-CoV-2 variant (s) and its impact on the prevalence of COVID-19 cases in the Nabatieh Region, Lebanon, which talk about the importance of lockdown in COVID-19 era, the emergence of SARS-CoV-2 variants and the increase of positive COVID-19 cases due to this variation).

3- Concerning the Materials and Methods section, there is no information in this section regarding how each type of interviewed persons was contacted neither the questions in his interview.

4- Concerning the Results section, this section is very long and contains a lot of information that must be used in the discussion. Authors are invited to make it more simple for readers.

5- Concerning the Discussion section, this section is very poor in information, authors are invited to discuss they results and compare it with data from different countries and from different regions in Argentina (International and National References must be used in this section).

Best Regards,

Reviewer 2 Report

The paper has been revised well. I only do not find the conclusion very convincing yet. You could draw out the connections between the different paper themes more clearly here.

Round 3

Reviewer 1 Report

Dear Authors,

The manuscript was re-reviewed.

I would like to thank you for the modifications.

The article is better in the present form.

BR,

Ghassan